# $\varepsilon$-fractional core stability in Hedonic Games

**Simone Fioravanti**[1]     **Michele Flammini**[1,2]     **Bojana Kodric** [3]     **Giovanna Varricchio**[2,4]

[1] Gran Sasso Science Institute (GSSI), L'Aquila, Italy
[2] University of Calabria, Rende, Italy
[3] Ca' Foscari University of Venice, Venice, Italy
[4] Goethe-Universität, Frankfurt am Main, Germany

{simone.fioravanti, michele.flammini}@gssi.it
bojana.kodric@unive.it   giovanna.varricchio@unical.it

## Abstract

Hedonic Games (HGs) are a classical framework modeling coalition formation of strategic agents guided by their individual preferences. According to these preferences, it is desirable that a coalition structure (i.e. a partition of agents into coalitions) satisfies some form of stability. The most well-known and natural of such notions is arguably core-stability. Informally, a partition is core-stable if no subset of agents would like to deviate by regrouping in a so-called core-blocking coalition. Unfortunately, core-stable partitions seldom exist and even when they do, it is often computationally intractable to find one. To circumvent these problems, we propose the notion of $\varepsilon$-fractional core-stability, where at most an $\varepsilon$-fraction of all possible coalitions is allowed to core-block. It turns out that such a relaxation may guarantee both existence and polynomial-time computation. Specifically, we design efficient algorithms returning an $\varepsilon$-fractional core-stable partition, with $\varepsilon$ exponentially decreasing in the number of agents, for two fundamental classes of HGs: Simple Fractional and Anonymous. From a probabilistic point of view, being the definition of $\varepsilon$-fractional core equivalent to requiring that uniformly sampled coalitions core-block with probability lower than $\varepsilon$, we further extend the definition to handle more complex sampling distributions. Along this line, when valuations have to be learned from samples in a PAC-learning fashion, we give positive and negative results on which distributions allow the efficient computation of outcomes that are $\varepsilon$-fractional core-stable with arbitrarily high confidence.

## 1 Introduction

Game-theoretic models of coalition formation have drawn significant interest in the last years, because of their ability to capture meaningful properties of multi-agent interactions. In Hedonic Games (HGs) [2, 16] agents gather together without any form of externality, that is only minding the internal composition of their groups. A solution is then a partition of the agents (or *coalition structure*) having some desirable properties, which typically stand for stability against some kinds of deviations. Among the many notions existing in the literature, one of the most fundamental is core stability [10, 31]. A partition is said to be *core-stable* or in the core if no subset of agents would benefit by regrouping and forming a so-called *core-blocking* coalition. Unfortunately, while being a quite natural requirement, it is notably very difficult to achieve [5, 25], even under the usual unrealistic assumption of full knowledge of agents' preferences. Furthermore, for the few classes of HGs in which the non-emptiness of the core has been established, a stable partition is usually hard to compute. Nonetheless, due to its significance, it is still desirable to find an approximation of the core.

37th Conference on Neural Information Processing Systems (NeurIPS 2023).

The core was first considered in the setting of cooperative game theory, where the value of a coalition has to be allocated fairly between its members. In this scenario, the most well-known approximation to the core is the so-called strong $\varepsilon$-core [26], in which a blocking coalition increases the total value allocated to its members by at least $\varepsilon$. A derived notion is the one of *least*-core [22], i.e. the (strong) $\varepsilon$-core associated to the smallest possible value of $\varepsilon$ guaranteeing its existence. An adaptation of these concepts to the context of HGs has been proposed in [17], where the authors define $k$-*improvement* core stability, requiring each member of a blocking coalition to increase her utility by a multiplicative factor strictly greater than $k \geq 1$. The same authors also propose to bound by a value $q \geq 2$ the number of agents allowed to form a blocking coalition, obtaining what they call $q$-*size* core stability.

However, the above stability concepts might still be fragile, especially when many coalitions could anyway benefit by deviating, even if not consistently according to the approximation/size factors. In fact, agents might be still inclined to subtle improvements. Interestingly, works like [11] experimentally show that, sampling random HGs, the fraction of instances with an empty core is small and decreases significantly as the number of agents increases. This could be related to the fact that, even in the instances with empty core, many or almost all the coalitions do not core-block. Consequently, the existence of outcomes having a relatively small number of blocking coalitions appears plausible.

In this work, we investigate the concept of $\varepsilon$-fractional core-stable partitions, i.e. partitions that can be core-blocked by only an $\varepsilon$-fraction of all possible coalitions. This notion bears some similarities with PAC-stabilizability as defined in [27]. In this work, the authors lifted for the first time the assumption of complete information in HGs and considered learning preferences and core-stable partitions from samples, employing the *probably approximately correct (PAC) learning* framework [30]. Specifically, an HGs class is PAC stabilizable if, after seeing a certain number of samples, it is possible to either determine that the core is empty or to return a partition that has probability at most $\varepsilon$ to be core-blocked by a coalition sampled from the same distribution.

**Our Contribution.**   In this paper, we introduce and study the $\varepsilon$-fractional core stability notion in Hedonic Games. We specifically investigate this concept on two of the most fundamental classes of HGs: *Simple fractional* and *anonymous* HGs.

Roughly speaking a partition of agents is $\varepsilon$-fractional core-stable, $\varepsilon$-FC in short, if at most an $\varepsilon$ fraction of coalitions may core-block it. Such a definition has a natural probabilistic interpretation: Indeed, for an $\varepsilon$-FC partition, $\varepsilon$ represents an upper bound to the probability of drawing a core-blocking coalition uniformly at random. Along this line, we broaden the definition to any distribution over coalitions by requiring that the probability of sampling a blocking coalition is at most $\varepsilon$. Worth noticing, if $\varepsilon = 0$, we are essentially requiring our solution to be core stable. Hence it is not possible in general to prove the existence and efficiently compute $\varepsilon$-FC outcomes, for values of $\varepsilon$ that are sufficiently close to 0. In contrast, our aim is to efficiently compute $\varepsilon$-FC solutions for as small as possible values of $\varepsilon$, proving in turn also their existence.

Unfortunately, as a first result, we prove that by allowing arbitrary sampling distributions an $\varepsilon$-FC may fail to exist for constant values of $\varepsilon$. On the positive side, for the aforementioned classes of HGs, we show that it is possible to efficiently compute an $\varepsilon$-FC solution under the uniform distribution, with $\varepsilon$ sub-exponentially small in the number of agents. In particular, in the case of anonymous HGs, we present an algorithm computing an $\varepsilon$-FC solution under the class of $\lambda$-*bounded* distributions, where the ratio of the probabilities of extracting any two coalitions is bounded by the parameter $\lambda$. Notably, this class includes the uniform distribution as the special case $\lambda = 1$, and can be considered a suitable extension of it. Our algorithms, besides guaranteeing sub-exponentially small values of $\varepsilon$, are designed to handle the possible incomplete knowledge of agents' preferences. In fact, in case preferences are unknown, the algorithms can use the sampling distribution to learn them in a PAC-learning fashion while maintaining the very same guarantees on $\varepsilon$ with high confidence.

## 2   Related Work

**Core stability and Hedonic Games.**   Hedonic Games have captured considerable research attention from the scientific community over the years. One of the main goals in their study is understanding how to gather agents in a way they won't desire to modify the outcome. For this reason, several stability notions have been considered and studied such as core and Nash stability, or individual rationality. We refer to [2] for a comprehensive overview of the subject. Core stability is a fundamental

concept in multi-agent systems, first considered in the context of cooperative games [13, 19]. Its properties and the related complexity have been largely investigated in HGs [1, 10, 25, 28] and beyond, such as in house allocation [24], markets [8, 9] and many other settings. Recently Donahue and Kleinberg [14, 15] have modeled federated learning as an HG, where agents evaluate federating coalitions according to the expected mean squared error of the model they obtain by sharing data with the coalition's members. Works like [4, 12] have used core stability to study payoff allocation among team members in collaborative multi-agent settings.

**PAC-stabilizability.** Our definition of $\varepsilon$-fractional core stability is also strictly related to PAC stabilizability as defined in [27]. This notion was further investigated in several papers. Igarashi et al. [20] studied the case of HGs with underlying interaction networks. Jha and Zick [21] defined a general framework for learning game-theoretic solution concepts from samples. Trivedi and Hemachandra [29] considered learning and stabilizing HGs with noisy preferences. Recently, Fioravanti et al. [18] proposed to relax the requirements of PAC-stabilizability by considering only restricted distributions and showed stabilizability of $\mathcal{W}$-games under $\lambda$-bounded distributions.

# 3 Preliminaries

In this section, we present the fundamental definitions for our work. Given a positive integer $k$ we use $[k]$ to denote the set $\{1, \ldots, k\}$.

## 3.1 Hedonic Games

Let $N$ be a set of $n$ *agents*. We call any non-empty subset $C \subseteq N$ a *coalition* and any coalition of size one a *singleton*. We denote by $\succsim_i$ any binary *preference relation* of agent $i$ over all coalitions containing $i$, which is reflexive, transitive, and complete. A Hedonic Game is then a pair $H = (N, \succsim)$, where $\succsim = (\succsim_i, \ldots, \succsim_n)$ is a *preference profile*, i.e., the collection of all agents' preferences.

Throughout this work, we will assume that preferences are expressed as real numbers by means of *valuation* functions $v_i : 2^N \to \mathbb{R}$ for each $i \in N$. In other words, given two coalitions $C, C'$ containing agent $i$, $v_i(C) \geq v_i(C')$ if and only if $C \succsim_i C'$. We will denote by $\vec{v} = (v_1, \ldots, v_n)$ the collections of agents' valuations and assume that $v_i$ is not defined for $C$ not containing $i$ and write $v_i(C) = \varnothing$. A *coalition structure* $\pi$ is a partition of agents into coalitions and $\pi(i)$ denotes the coalition $i$ is assigned to. We write $v_i(\pi) = v_i(\pi(i))$ to denote the utility $i$ gets in her assigned coalition $\pi(i)$ inside $\pi$.

With this paper, we aim to relax the concept of core stability defined as follows.

**Definition 3.1.** *Given a coalition structure $\pi$, a coalition $C \subseteq N$ is said to* core-block *$\pi$ if, for each $i \in C$, $v_i(C) > v_i(\pi)$. A coalition structure $\pi$ is said to be* core-stable *if no coalition $C \subseteq N$ core-blocks it.*

**Simple fractional Hedonic Games.** In *fractional* HGs (FHGs), first introduced in [3], every agent $i \in N$ assigns a value $v_i(j)$ to any other $j \neq i$, and then her evaluation for any coalition $C \ni i$ is the average value ascribed to the members of $C \setminus \{i\}$. Formally: $v_i(C) = \frac{\sum_{j \in C \setminus \{i\}} v_i(j)}{|C|}$. A FHG is said to be *simple* if $v_i(j) \in \{0, 1\}$, for each $i, j \in N, i \neq j$. A natural representation of these games is a directed and unweighted graph $G = (V, E)$, where $V = N$ and $(i, j) \in E$ if and only if $v_i(j) = 1$. Despite their name, Aziz et al. [3] show that simple FHGs capture the complexity of the entire class w.r.t. core-stability: In fact, deciding if a core-stable partition exists is $\Sigma_2^p$-complete.

**Anonymous Hedonic Games.** An HG is said to satisfy *anonimity* [6, 10] if agents evaluate coalitions on the sole basis of their size, i.e., $v_i(C) = v_i(C')$ for any $i \in N$ and any $C, C'$ containing $i$ such that $|C| = |C'|$. When considering anonymous HGs we will assume $v_i : [n] \to \mathbb{R}$. An anonymous HGs instance is said to be *single-peaked* if there exists a permutation $(s_1, \ldots, s_n)$ of $\{1, \ldots, n\}$ for which every agent $i \in N$ admits a *peak* $p(i) \in [n]$ such that $h < k \leq p(i)$ or $h > k \geq p(i)$ imply $v_i(s_k) \geq v_i(s_h)$. Roughly speaking, the higher the distance from the peak in the given ordering, the lower the valuation for the coalition size. If the permutation is the identity function, i.e. the ordering is the usual one over $\mathbb{N}$, we say that the preference is *single-peaked in the*

*natural ordering*. For anonymous HGs, deciding if a core-stable partition exists has been shown to be NP-complete [5].

## 3.2 Epsilon-core, learning, and computation efficiency

Here we formalize our notion of $\varepsilon$-fractional core stability.

**Definition 3.2.** *Given a parameter $\varepsilon \in [0,1]$, we say that a partition $\pi$ is $\varepsilon$-fractional core-stable, $\varepsilon$-FC in short, if it holds that at most an $\varepsilon$-fraction of all possible coalitions are core-blocking for partition $\pi$, i.e.,*

$$\frac{\text{\# of core-blocking coalitions for } \pi}{\text{\# of all possible coalitions}} < \varepsilon .$$

This definition has a natural probabilistic interpretation: A partition is $\varepsilon$-FC if, by sampling u.a.r. a coalition, the probability of sampling a core-blocking one is at most $\varepsilon$. Such an interpretation inspired the following extension.

**Definition 3.3.** *Given a parameter $\varepsilon \in [0,1]$, we say that a partition $\pi$ is $\varepsilon$-fractional core-stable with respect to a distribution $\mathcal{D}$ over $2^N$ if*

$$\Pr_{C \sim \mathcal{D}}[C \text{ core blocking for } \pi] < \varepsilon .$$

We may assume that agents' preferences are unknown and we need to learn them by sampling coalitions from a distribution $\mathcal{D}$. Consequently, our algorithms will have a *learning phase*, where preferences are learned by observing $m$ samples, and a *computation phase*, where a stable outcome is computed upon the learned preferences. We say that an algorithm *exactly learns* a family $\mathcal{T} \subseteq 2^N$ if after seeing a sample $\mathcal{S}$ it is able to determine the real valuation of any agent for every coalition in $\mathcal{T}$. Note that this does not necessarily mean that $\mathcal{T} \subseteq \mathcal{S}$; instead, by knowing the properties of the considered HG class, it must be possible to derive complete information of $\mathcal{T}$ from $\mathcal{S}$. As an example, consider the anonymous HGs class: If $\mathcal{T}$ is the family of coalitions of size $s$, in order to exactly learn it, for each agent $i$ there must exist $S \in \mathcal{S}$ such that $i \in S$ and $|S| = s$.

We aim to design polynomial-time algorithms for computing $\varepsilon$-FC solutions. However, to retrieve enough information on the preferences we may require a large number of samples. Hence, we say that an algorithm is *efficient* if its computation phase requires polynomial time in the instance size while the learning phase is polynomial in $n, 1/\varepsilon, \log 1/\delta$, where $\delta$ is a confidence parameter. Clearly, if the valuations are known in advance, an efficient algorithm requires polynomial time. Moreover, our algorithms will compute an $\varepsilon$-FC partition with confidence $1 - \delta$ and the solution will turn out to be exact as soon as the true agents' preferences are given as input.

## 3.3 Chernoff Bound and $\lambda$-bounded distributions

The analysis of our algorithms is mainly probabilistic and will strongly rely on the classical Chernoff bound (see, e.g., Chapter 4 in [23]) that we summarize hereafter. Let $X = \sum_{i=1}^{n} X_i$ be a sum of independent Poisson trials of mean $\mu$, then for any constant $b \in (0,1)$ it holds:

$$\Pr[X \geq (1+b)\mu] \leq e^{-\mu b^2/3} \qquad \text{and} \qquad \Pr[X \geq (1-b)\mu] \leq e^{-\mu b^2/2} . \qquad (1)$$

As we already mentioned, we will study the $\varepsilon$-fractional core stability subject to distributions $\mathcal{D}$ over $2^N$. We will see that, while for general distributions it is not possible to guarantee good enough values of $\varepsilon$, it is indeed possible for $\lambda$-*bounded* distributions. This class of distributions has been first introduced in a learning context by [7], where they consider general continuous distributions over bounded subsets of $\mathbb{R}^d$. The idea is that a distribution in this class has constraints (parameterized by the value $\lambda$) on how much the probability density function can vary.

**Definition 3.4.** *A distribution $\mathcal{D}$ over $2^N$ is said to be $\lambda$-bounded, if there exists $\lambda \geq 1$ such that, for every two coalitions $C_1, C_2$, it holds that*

$$\Pr_{C \sim \mathcal{D}}[C = C_1] \leq \lambda \Pr_{C \sim \mathcal{D}}[C = C_2] .$$

A straightforward consequence of this definition is that no coalition has null probability of being sampled. Moreover, it can be noted that, while setting $\lambda = 1$ we obtain the uniform distribution

over $2^N$, as $\lambda \to +\infty$ every distribution can be considered $\lambda$-bounded up to an approximation factor. Thus, in order to keep the original intended purpose of this definition, in the rest of this work we will consider $\lambda$ to be constant with respect to $n$.

The following lemma, an adaptation of Lemma 4 in [7], will be useful in our computations.

**Lemma 3.5.** *Let distribution $\mathcal{D}$ be $\lambda$-bounded. Let $\mathcal{F} \subseteq \mathcal{P}(2^N)$ be a family of subsets and let $a = |\mathcal{F}|/2^n$. Then, the following inequalities hold:*

$$\frac{a}{a + \lambda(1 - a)} \leq \Pr_{C \sim \mathcal{D}}[C \in \mathcal{F}] \leq \frac{\lambda a}{\lambda a + 1 - a}.$$

# 4 Impossibility results

In this section, we explore the boundaries of feasible $\varepsilon$ for simple fractional and anonymous HGs according to general, $\lambda$-bounded, and uniform distributions.

**Simple fractional Hedonic Games.** We start by showing that when dealing with arbitrary distributions, an $\varepsilon$-FC may not exist for $\varepsilon$ exponentially small w.r.t. the number of agents. Specifically, this impossibility result holds for constant values of $\varepsilon$. The informal intuition is that, being the distribution arbitrary, one may choose it in an adversarial way with respect to a partition having an empty core.

**Proposition 4.1.** *There exists a distribution $\mathcal{D}$ and a simple fractional HG instance such that no $\varepsilon$-fractional core-stable solution w.r.t. $\mathcal{D}$ exists for $\varepsilon \leq 1/2^{40}$.*

*Proof.* Simple fractional HGs have been shown to have an empty core [3]. In particular, the authors provide an instance $\mathcal{I}$ with 40 agents having an empty core. We can extend this instance to an instance $\mathcal{I}'$, with $N' = [n]$ being the set of agents, still having an empty core. Let $N = \{1, \ldots, 40\}$ be the set of agents in $\mathcal{I}$. Their mutual preferences remain the same, while they value 0 all the other agents in $N' \setminus N$. In turn, the agents in $N' \setminus N$ have mutual preferences equal to 1, and they value 0 all the agents in $N$. $\mathcal{I}'$ has an empty core and, in particular, for any coalition structure $\pi$ there exists a core blocking coalition in $2^N \cup \{\{41, \ldots, n\}\} \setminus \{\emptyset\}$. In fact, on the one hand, the agents in $\{41, \ldots, n\}$ will form a blocking coalition whenever we return a partition where they are not in the same coalition. On the other hand, if $\{41, \ldots, n\}$ is a coalition of the considered partition, no matter how the agents in $N$ will be partitioned, there exists a blocking coalition in $2^N \setminus \{\emptyset\}$ because the instance $\mathcal{I}$ has empty core. In conclusion, by choosing $\mathcal{D}$ as the uniform distribution over $2^{N'} \cup \{\{41, \ldots, n\}\} \setminus \{\emptyset\}$, for any coalition structure $\pi$ we have $\Pr_{C \sim \mathcal{D}}[C$ blocking for $\pi] > 1/2^{40}$ and the thesis follows. $\square$

Similarly, by applying Lemma 3.5, we can easily derive the following generalization.

**Corollary 4.2.** *Given a parameter $\lambda$ there exists a bounded distribution with parameter $\lambda$ such that in simple fractional HGs no $\varepsilon$-core exists for $\varepsilon < \frac{\lambda}{2^{40}(\lambda-1)+2^n}$.*

**Anonymous Hedonic Games.** In the following, we consider single-peaked anonymous HGs. Clearly, all the provided results hold for the more general class of anonymous HGs. Following the same approach of the previous section, knowing that for anonymous HGs there exists an instance with seven agents and single-peaked preferences having empty core [6], we can show the following.

**Proposition 4.3.** *There exists a distribution $\mathcal{D}$ and an anonymous (single-peaked) HG instance such that for every $\varepsilon \leq 1/2^7$ there is no $\varepsilon$-fractional core w.r.t. $\mathcal{D}$.*

**Corollary 4.4.** *Given a parameter $\lambda$, there exists a $\lambda$-bounded distribution such that for $\varepsilon < \frac{\lambda}{2^7(\lambda-1)+2^n}$ no $\varepsilon$-fractional core-stable solution exists in anonymous single-peaked HGs.*

As a consequence, both for simple fractional and anonymous HGs, under uniform distributions, where $\lambda = 1$, and constant values of $\lambda$, the provided bound becomes exponentially small. Hence, in the rest of this paper, we will be focusing on these specific classes of distributions.

# 5 Simple Fractional Hedonic Games

In this section, we present an algorithm returning an $\varepsilon$-FC for simple FHGs with $\varepsilon$ exponentially small in the number of agents, and prove its correctness, as summarized in the following claim.

**Theorem 5.1.** *Given a parameter $\delta \in (0, 1)$, for any simple FHG instance and with confidence $1 - \delta$, we can efficiently compute an $\varepsilon$-fractional core-stable partition for every $\varepsilon \geq 2^{-\Omega(n^{1/3})}$.*

### 5.1 Computing an $\varepsilon$-fractional core-stable partition

We will be working only with the uniform distribution over all possible coalitions denoted by $U(2^N)$.

The proof of the theorem is quite involved and needs several steps. To facilitate the reader, the section is organized as follows: First, we provide a high-level explanation of Algorithm 1 for computing an $\varepsilon$-FC partition, then we show various parts of the proof as separate lemmas, and finally, we assemble all together to prove Theorem 5.1.

As a first step, Algorithm 1 consists of a learning phase where the exact valuations are computed with confidence $1 - \delta$. Then, it starts the computation phase building the graph representation $G$ based on the learned preferences. In order to compute an $\varepsilon$-FC partition, the algorithm has to pay attention to the density of $G$. In particular, the main intuition is to distinguish between instances based on the number of nodes having high out-degrees, treating the two arising cases separately. To this aim, observe first that the biggest fraction of (sampled) coalitions has a size close to $\frac{n}{2}$[1]. As a consequence, if many nodes have low out-degree, a matching-like partition would be difficult to core-block, since a sampled coalition $C$ will often contain very few neighbors of such nodes, compared to its size. Thus, such nodes would have a lower utility in $C$. On the other hand, if a sufficient amount of nodes have a high out-degree, we can form a clique large enough to make any partition containing it hard to core-block. In fact, with high probability, the sampled coalition $C$ would contain at least one of such clique nodes, not connected to all the other agents in $C$. Therefore, it would have a lower utility moving to $C$.

First, we focus on the learning aspect and show that it is possible to learn the exact valuations in simple FHGs, upon sampling a number of sets polynomial in $n$ and $\log \frac{1}{\delta}$.

**Lemma 5.2.** *By sampling $m \geq 16 \log \frac{n}{\delta} + 4n$ sets from $U(2^N)$ it is possible to learn exactly the valuation functions $v_1, \dots, v_n$ with confidence $1 - \delta$.*

The following definition is crucial for proving the main result.

**Definition 5.3.** *Given a partition $\pi$, an agent $i \in N$ is* green *with respect to $\pi$ if*

$$\Pr_{C \sim U(2^N)} [v_i(C) > v_i(\pi) | i \in C] \leq 2^{-\Omega(n^{1/3})} . \tag{2}$$

Since a coalition containing a green agent w.r.t. $\pi$ is unlikely to core-block $\pi$, if we manage to show that there are enough green agents in the partition returned by Algorithm 1, then we directly prove also its $\varepsilon$-fractional core-stability, as in this case any randomly drawn coalition contains at least one green agent with high probability. From now on, let $\phi = |\{i \text{ s.t. } d_i \leq n - 31n^{2/3}\}|$ be as defined in Algorithm 1. As already informally explained above, the algorithm follows two different procedures based on $\phi$ being greater equal or lower than $\frac{n^{1/3}}{62}$. The set Gr defined in Algorithm 1 in both cases is meant to contain exclusively green agents w.r.t. the returned partition. We will start by proving that this is indeed true if $\phi \leq \frac{n^{1/3}}{62}$. Since we will never use in-degrees, from now on we will simply use the term *degree* in place of out-degree.

We first prove the correctness of the easier case of the algorithm in the else part of the if statement (line 19).

**Lemma 5.4.** *Let $\phi < \frac{n^{1/3}}{62}$ and $\pi$ be the partition returned by Algorithm 1. Then Gr contains only green agents w.r.t to $\pi$.*

*Proof.* Let us say that an agent $i \in N$ has *high* degree if $d_i \geq n - 31n^{2/3}$. Observe that, if such an agent is picked at any iteration of the algorithm, then at most $31n^{2/3}$ agents are deleted from $F$ at that iteration. By hypothesis, there are at least $n - \frac{n^{1/3}}{62}$ agents with high degree. Since they are picked in non-increasing degree order, after $t$ iterations the agents left in $F$ with high degree are at least $n - \frac{n^{1/3}}{62} - t \cdot 31n^{2/3}$, which is strictly positive for $t \leq \frac{n^{1/3}}{124}$. This means that all agents in Gr

---

[1] Note that $\frac{n}{2}$ is the average size of coalitions drawn from the uniform distribution.

---
**Algorithm 1:** Stabilizing Simple FHGs
---
**Input:** $N, \mathcal{S} = \langle (S_j, \vec{v}(S_j)) \rangle_{j=1}^m$

**Output:** $\pi$ – a $2^{-\Omega(n^{1/3})}$-fractional core-stable partition of $N$

1   Compute $\vec{v}$ and build the corresponding graph $G$

2   $\pi \leftarrow \{\{i\} | i \in N\}$

3   Let $d_i$ be the out-degree of each $i$ in $G$

4   $\phi \leftarrow |\{i \text{ s.t. } d_i \leq n - 31n^{2/3}\}|$

5   $\mathrm{Gr} \leftarrow \emptyset$

6   **if** $\phi \geq \frac{n^{1/3}}{62}$ **then**

7      Sort agents in non-decreasing order of out-degree

8      $H \leftarrow \left[ 1, \ldots, \frac{n^{1/3}}{62} \right]$

9      $count \leftarrow 1$

10      **while** $count \leq \frac{n^{1/3}}{124}$ **do**

11         Select first remaining agent $i \in H$ in the order

12         $\mathrm{Gr} \leftarrow \mathrm{Gr} \cup \{i\}$

13         Let $F_i$ be a set of $\left\lceil \frac{2d_i}{n-d_i} \right\rceil$ neighbors of $i$, giving priority to singleton agents in $N \setminus H$

14         Let $C_i = \{i\} \cup \bigcup_{j \in F_i} \pi(j)$

15         $\pi \leftarrow \pi \cup \{C_i\} \setminus \{\pi(j) | j \in C_i\}$

16         $H \leftarrow H \setminus (F_i \cup \{i\})$

17         $count \leftarrow count + 1$

18      All remaining singletons $\pi(i) = \{i\}$ in $\pi$ are grouped together

19   **else**

20      $F \leftarrow N$

21      **for** $count = 1, \ldots, \frac{n^{1/3}}{124}$ **do**

22         Pick $i \in F \setminus \mathrm{Gr}$ of maximal out-degree

23         Delete from $F$ all agents not in the neighborhood $N_i$ of $i$

24         $\mathrm{Gr} \leftarrow \mathrm{Gr} \cup \{i\}$

25      $\pi \leftarrow \{F, N \setminus F\}$

26   **return** $\pi$

---

have high degree, so that $|F| \geq n - 31n^{2/3} \cdot \frac{n^{1/3}}{124} = \frac{3n}{4}$. Moreover, since by construction at the end of the for loop each $i \in \mathrm{Gr}$ is connected to all the other agents in $F$, $v_i(F) \geq 1 - \frac{4}{3n}$, which is at least the utility $i$ would have in a clique of $\geq \frac{3n}{4}$ agents. Therefore, any coalition of size at most $\frac{3n}{4}$ containing $i$ cannot core block $\pi$. As a consequence, for each $i \in \mathrm{Gr}$ it holds that:

$$\Pr_{C \sim U(2^N)} \left[ v_i(C) > v_i(F) | i \in C \right] \leq \Pr_{C \sim U(2^N)} \left[ |C| > \frac{3n}{4} \Big| i \in C \right] \leq e^{-\frac{n}{24}} < 2^{-\Omega(n^{1/3})} ,$$

where we used Equation (1) and the fact that $n > 1$.        $\square$

The procedure returning a partition in the complementary case $\phi \geq \frac{n^{1/3}}{62}$ is a bit more involved, and we divide the proof in different cases according to the degree of the agents added to Gr.

**Lemma 5.5.** *Let $\phi \geq \frac{n^{1/3}}{62}$ and let $\pi$ be the partition returned by Algorithm 1 in this case. Then the following statements hold:*

     *(i) $H$ is never empty when executing line 11 of the while loop of Algorithm 1;*

*(ii) each agent $i \in \mathrm{Gr}$ is green.*

*Proof of Theorem 5.1.* We are now able to assemble all the above claims together to prove the main theorem. According to the above lemmas, in both the cases of Algorithm 1, i.e. $\phi < \frac{n^{1/3}}{62}$ and $\phi \geq \frac{n^{1/3}}{62}$, the nodes put in Gr are green and $\gamma := |\mathrm{Gr}| = n^{1/3}/124$.

It remains to show that the output $\pi$ of Algorithm 1 is a $2^{-\Omega(n^{1/3})}$-fractional core-stable partition:

$$\Pr_{C \sim U(2^N)} \left[ C \text{ blocks } \pi \right] \leq \Pr_{C \sim U(2^N)} \left[ C \cap \mathrm{Gr} = \emptyset \right] + \Pr_{C \sim U(2^N)} \left[ C \cap \mathrm{Gr} \neq \emptyset \wedge C \text{ blocks } \pi \right]$$

$$\leq \frac{2^{n-\gamma}}{2^n} + \left(1 - \frac{2^{n-\gamma}}{2^n}\right) \frac{1}{2^{n^{1/3}}}$$

$$\leq \frac{1}{2^\gamma} + \frac{1}{2^{n^{1/3}}} \leq \frac{1}{2^{\frac{n^{1/3}}{124}-1}}.$$

and that concludes the proof. $\square$

# 6 Anonymous Hedonic Games

In this section, we discuss the efficient computation of an $\varepsilon$-FC partition in the case of anonymous HGs, and focus on the more general class of $\lambda$-bounded distributions.

The main result of this section is given by the following:

**Theorem 6.1.** *Given a $\lambda$-bounded distribution $\mathcal{D}$ and a parameter $\delta \in (0,1)$, for any anonymous HG instance and with confidence $1 - \delta$, we can efficiently compute an $\varepsilon$-fractional core-stable partition for every $\varepsilon \geq \frac{4\lambda}{2^{c(\lambda)\sqrt[3]{n}}}$, where $c(\lambda) = \frac{1}{\sqrt{13(\lambda+1)}}$.*

Moreover, when agents' preferences are single-peaked, we can refine the bound on $\varepsilon$ as follows.

**Theorem 6.2.** *Given a $\lambda$-bounded distribution $\mathcal{D}$ and a parameter $\delta \in (0,1)$, for any single-peaked anonymous HG instance and with confidence $1 - \delta$, we can efficiently compute an $\varepsilon$-fractional core-stable partition for every $\varepsilon \geq 4 \cdot \frac{\lambda}{2^{n/4}}$.*

## 6.1 Distribution over coalitions sizes

Being anonymous preferences uniquely determined by coalition sizes, it is important to establish how the $\lambda$-bounded distribution $\mathcal{D}$ impacts the probability of sampling coalitions of a given size.

Let $X : 2^N \to [n] \cup \{\emptyset\}$ be the random variable corresponding to the size of $C \sim \mathcal{D}$, i.e. $X(C) := |C|$, and $\mu := E[X]$ be its average. The following lemma shows some useful properties of $X$.

**Lemma 6.3.** *Let $X$ be the random variable representing the size of $C \sim \mathcal{D}$, with $\mathcal{D}$ $\lambda$-bounded, and let $\mu := E[X]$. Then,*

$$\frac{n}{\lambda+1} \leq \mu \leq \frac{\lambda n}{\lambda+1} \qquad \text{and} \qquad \Pr_{\mathcal{D}}\left[ |X - \mu| \geq \Delta \cdot \mu \right] \leq \frac{\varepsilon}{2},$$

*where $\varepsilon$ is such that $0 < \varepsilon < 1$ and $\Delta$ is the quantity $\sqrt{\frac{3(\lambda+1)\log\frac{4}{\varepsilon}}{n}}$.*

Let us denote by $I_\mathcal{D}(\varepsilon) \subseteq [n]$ the open interval $((1-\Delta)\mu, (1+\Delta)\mu)$, where $\mu$ is the expected value for the size of a coalition sampled from $\mathcal{D}$. Lemma 6.3 implies that with probability at least $1 - \varepsilon/2$ we draw from $\mathcal{D}$ a coalition whose size is in $I_\mathcal{D}(\varepsilon)$.

## 6.2 Properties of $I_\mathcal{D}(\varepsilon)$

The interval $I_\mathcal{D}(\varepsilon)$ will play a central role in the computation of an $\varepsilon$-FC partition. However, under the uncertainty of a distribution $\mathcal{D}$, our algorithms need to i) estimate it (in fact, $I_\mathcal{D}(\varepsilon)$ is uniquely determined by the value $\mu$ which is unknown) and ii) learn exactly the agents' valuations for coalitions whose size is in $I_\mathcal{D}(\varepsilon)$. The technical proofs of this subsection are deferred to the Appendix.

Being $\mu$ unknown, we will work on a superset of $I_{\mathcal{D}}(\varepsilon)$ which can be estimated by simply knowing that $\mathcal{D}$ is $\lambda$-bounded. Let $\mathcal{S} = \{S_j\}_{j=1}^{m}$ be a sample of size $m$ drawn from $\mathcal{D}$, and let $\bar{\mu} = \frac{1}{m} \sum_j |S_j|$ be the frequency estimator. By the Hoeffding bound (see Theorem 4.13 in [23]), it is possible to show that we can estimate $\mu$ with high confidence, as stated in the following.

**Lemma 6.4.** *Given any two constants $\alpha > 0, \delta < 1$, if $m > \frac{n^2 \log 2/\delta}{2\varepsilon^2}$, then:*

$$\Pr_{\mathcal{S} \sim \mathcal{D}^m} \left[ |\bar{\mu} - \mu| < \alpha \right] \geq 1 - \delta \;. \tag{3}$$

As a consequence, we can determine a good superset of $I_{\mathcal{D}}(\varepsilon)$ as the interval with extreme points $(1 \pm \Delta)(\bar{\mu} \pm \alpha)$.

We now turn our attention to the exact learning of $I_{\mathcal{D}}(\varepsilon)$.

**Lemma 6.5.** *By sampling $m = \frac{2\lambda(1+\lambda)n^2 \log n^2/\delta}{\varepsilon}$ sets from $\mathcal{D}$ it is possible to learn exactly the valuations in $I_{\mathcal{D}}(\varepsilon)$, with confidence $1 - \delta$.*

### 6.3 Computing an $\varepsilon$-fractional core-stable partition for bounded distributions

Our algorithm will consist of a learning phase and a computation phase.

During the learning phase, the algorithm passes through a sample of size $m = \frac{2\lambda(1+\lambda)n^2 \log n^2/\delta}{\varepsilon}$ and for each coalition $C$ in the sample of size $c$ and $i \in C$ it stores the value $v_i(c)$. Let us denote by $\mathcal{X}$ the coalition sizes that have been learned exactly during this learning phase, that is, the coalition sizes for which the algorithm learned the valuations of each agent. By the previous lemma, with confidence $1 - \delta$, $I_{\mathcal{D}}(\varepsilon) \subseteq \mathcal{X}$.

During the computation phase, our algorithm will make sure that a sufficiently large amount of agents have a very small probability of being involved in a core blocking coalition. Such agents will be called green agents and are defined as follows. An agent $i$ is said to be *green* in a partition $\pi$ w.r.t. $I \subseteq [n]$, if $|\pi(i)|$ maximizes $v_i(s)$ for $s \in I$. We will show that if the probability of sampling coalitions of sizes $s \in I$ is high enough, then green agents do not want to deviate from their actual partition with high probability as well. As a consequence, a partition containing many green agents is difficult to core-block, as shown in the following:

**Lemma 6.6.** *For any partition $\pi$ of agents and $I \subseteq [n]$ such that $I_{\mathcal{D}}(\varepsilon) \subseteq I$. If $\pi$ contains at least $\log_2 \frac{2\lambda}{\varepsilon}$ green agents w.r.t. $I$, then, $\pi$ is $\varepsilon$-fractional core-stable.*

*Proof.* Let $C \sim \mathcal{D}$ and $c = |C|$ be its size. We denote by $\mathcal{E}_1$ the event that $C$ core blocks $\pi$, and by $\mathcal{E}_2$ then the event that $C$ does not contain green agents. By definition of green agents, if $c \in I$ and $C$ contains at least one green agent w.r.t. $I$, then it cannot core block $\pi$. Formally speaking, $\overline{\mathcal{E}_2} \wedge c \in I \Rightarrow \overline{\mathcal{E}_1}$ and therefore $\mathcal{E}_1 \Rightarrow \mathcal{E}_2 \vee c \notin I$. As a consequence,

$$\Pr_{\mathcal{D}} \left[ \mathcal{E}_1 \right] \leq \Pr_{\mathcal{D}} \left[ c \notin I \vee \mathcal{E}_2 \right] \leq \Pr_{\mathcal{D}} \left[ c \notin I \right] + \Pr_{\mathcal{D}} \left[ \mathcal{E}_2 \right] \;.$$

Being $I_{\mathcal{D}}(\varepsilon) \subseteq I$, by Lemma 6.3 and Lemma 3.5 we finally get $\Pr_{\mathcal{D}} \left[ \mathcal{E}_1 \right] \leq \frac{\varepsilon}{2} + \lambda \frac{2^{n - \log_2 \frac{2\lambda}{\varepsilon}}}{2^n} = \varepsilon$. $\square$

We finally sketch the proof of Theorem 6.1, for a rigorous proof see the Appendix. The key idea is to create a partition $\pi$ by grouping as many agents as possible in a coalition of preferred size among a set of sizes, say $I$, that may occur with particularly high probability. Hence, by Lemma 6.6 we only need to find appropriate $I$ and $\pi$ such that the number of green agents for $\pi$ w.r.t. $I$ is at least $\log_2 \frac{2\lambda}{\varepsilon}$.

*Proof of Theorem 6.1.* Let us start by defining the set $I$. By Lemma 6.4, we can provide an estimation $\bar{\mu}$ of $\mu$ such that $\mu \in (\bar{\mu} - \alpha, \bar{\mu} + \alpha)$ for a certain $\alpha$ with confidence $1 - \delta$. As a consequence, if we can consider $I$ as the intersection of the interval having extreme points $(1 \pm \Delta)(\bar{\mu} \pm \alpha)$ and the set $\mathcal{X}$ of sizes exactly learned during the learning phase. By Lemma 6.5, with confidence $1 - \delta$, we can say that $I_{\mathcal{D}}(\varepsilon) \subseteq \mathcal{X}$ since with such confidence $I_{\mathcal{D}}(\varepsilon)$ has been learned. By union bound, with confidence $1 - 2\delta$, $I_{\mathcal{D}}(\varepsilon) \subseteq I$. W.l.o.g. we can assume our confidence to be $1 - \delta$ by replacing $\delta$

with $\delta/2$. Moreover, being $|I| \leq 2(\Delta\bar{\mu} + \alpha)$ and $\bar{\mu} \geq 1$ and choosing $\alpha = \min\{\frac{1}{2\sqrt{n}}, \frac{n}{\lambda+1}\}$ we can derive the following bound:

$$|I| \leq n\sqrt{\frac{13(\lambda+1)\log\frac{4}{\varepsilon}}{n}}.$$

By the pigeonhole principle, there is one $s \in I$ such that at least $\frac{n}{|I|}$ agents prefer $s$ over the other coalition sizes in $I$. Using the hypothesis on $\varepsilon$, we can conclude $\frac{n}{|I|} \geq \log_2\frac{2\lambda}{\varepsilon}$.

Let us now create the desired $\pi$. Let $q, r$ be two positive integers s.t. $n = qs + r$ and $r < s$. Giving priorities to the agents having $s$ as the most preferred size in $I$, we create $q$ coalitions of size $s$ and a coalition of size $r$. In this way, at least $\log_2\frac{2\lambda}{\varepsilon}$ agents are in a coalition having the most preferred size within $I$ and, therefore, they are green agents for $\pi$ w.r.t. $I$. By Lemma 6.6, the thesis follows. □

Finally, we turn our attention to a special but really popular subclass of anonymous HGs. In this case, thanks to the further assumption of single-peakedness of agents' preferences we can achieve better values of $\varepsilon$. Because of space constraints, the proof of Theorem 6.2 is deferred to the Appendix.

## 7 Conclusions and Future Work

We introduced and studied the $\varepsilon$-fractional core stability concept, a natural relaxation of core stability where only a small ratio (or a small probability mass) of coalitions is allowed to core-block.

We investigated this concept on two fundamental classes of HGs: Simple FHGs and anonymous HGs. For both these classes the problem of deciding the existence of a core-stable partition is notably hard: NP-complete for anonymous HGs and even $\Sigma_2^p$- complete for simple FHGs. While in Section 4 we showed that very small value of $\varepsilon$ and the choice of the distributions pose limits to the existence of $\varepsilon$-FC solutions, we have still been able to obtain positive results for both classes under different assumptions on the considered distributions. For simple FHGs we showed that, when sampling from a uniform distribution (that corresponds to $\varepsilon$-FC as in Definition 3.2), it is possible to always construct an $\varepsilon$-FC under the assumption that $\varepsilon$ does not decrease faster than a sub-exponential value. For anonymous HGs instead, we were able to show the existence of $\varepsilon$-FC for a much broader class of distributions (i.e. the $\lambda$-bounded ones), under a similar condition over $\varepsilon$. These encouraging results show that, while having a natural probabilistic nature, the notion of $\varepsilon$-FC stability is flexible enough to allow positive results in very complex settings where core-stability is usually considered unattainable. Moreover, its very definition and connection with the concept of PAC stabilizability makes it resilient to possible uncertainty of agents' preferences, increasing its usability in applications.

**Limitations of $\varepsilon$-fractional core stability.** Beyond the core, many stability concepts have been introduced and studied in the literature of HGs. Among the others, we mention individual rationality (IR) which is considered the minimum requirement of stability: It postulates that no agent strictly prefers being alone, forming a singleton, rather than being in their current coalition. Clearly, core stability implies IR; in fact, if the outcome is not IR there must exist an agent able to form a blocking coalition on her own. Despite IR being implied by core-stability, this is not the case for $\varepsilon$-FC and the algorithms presented in this paper do not necessarily satisfy IR. Specifically, in the case of simple fractional HGs, any outcome is IR and therefore it is our proposed solution; on the other hand, in the case of anonymous HGs, our algorithm provides an IR solution if coalitions of size 1 are less preferred than any other coalition size. That said, we believe it is possible to modify our algorithm, under the reasonable assumption that the value of the singleton coalition is known for each agent, in a way that also IR is satisfied, at the cost of a possible worse lower bound on $\varepsilon$.

**Future directions.** This contribution has a high potential for future work. A first natural question is whether it is possible to extend our positive results for simple FHGs to the more general class of $\lambda$-bounded distributions. We believe that such an extension is attainable but nonetheless non-trivial. Moreover, although we showed that exponentially small values of $\varepsilon$ are not possible, it is still worth investigating which is the best guarantee for general distributions. More broadly, there are several HG classes that have not been considered in our work, and understanding for which $\varepsilon$ an $\varepsilon$-FC partition exists is certainly of interest. As discussed in the previous paragraph, it would be definitely of interest to come up with an algorithm for anonymous HGs guaranteeing the solutions to be IR. Generally speaking, it would be worth studying $\varepsilon$-FC in conjunction with IR in other HGs classes.

## Acknowledgements

We acknowledge the support of the PNRR MIUR project FAIR - Future AI Research (PE00000013), Spoke 9 - Green-aware AI, the PNRR MIUR project VITALITY (ECS00000041), Spoke 2 ASTRA - Advanced Space Technologies and Research Alliance, the Italian MIUR PRIN 2017 project AL-GADIMAR - Algorithms, Games, and Digital Markets (2017R9FHSR_002) and the DFG, German Research Foundation, grant (Ho 3831/5-1).

We thank the anonymous reviewers for their insightful comments and suggestions, which helped us to improve the quality of the manuscript.

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
