# OpenReview forum: "$\varepsilon$-fractional core stability in Hedonic Games."
_NeurIPS.cc/2023/Conference — NeurIPS 2023 poster_

### Official Review · Reviewer_psiE · 2023-06-20

**Soundness:** 3 good
**Presentation:** 3 good
**Contribution:** 3 good
**Rating:** 6
**Confidence:** 4

**Summary:**

This paper introduces the concept of an $\epsilon$-fractional core stability, relaxing the standard notion of core stability by allowing at most an $\epsilon$ fraction of all possible outcomes to core-block. They show that for sufficiently small values of $\epsilon$, $\epsilon$-fractional core stability always exists and can be computed in polynomial time for certain important subclasses of hedonic games, bypassing well-known difficulties with the usual notion of core stability. They also provide learning algorithms with PAC-type guarantees for identifying $\epsilon$-fractional core-stable outcomes under a broad class of underlying distributions over coalitions, as well as negative results for arbitrary distributions.

**Strengths:**

The paper makes a concrete contribution in the algorithmic study of core-stability notions in important classes of hedonic games, namely simple fractional and anonymous hedonic games. More precisely, the usual notion of core stability is known to suffer from two major drawbacks: it is not universal, and even when it exists it can be hard compute. As a result, different relaxations have been studied over the years that try to address those issues. This paper lies in this line of work, proposing a natural relaxation that only requires most of the coalitions to not be core-blocking. Moreover, they provide a number of non-trivial positive and negative results under this new concept. As such, I believe it makes concrete contributions in this line of work, the results have been nicely placed into the existing literature, and the paper can bring much interesting future work.

Moreover, the paper is well-written and structured, and all the important ideas and techniques and sufficiently exposed in the main body. I did not find any notable issue in the technical part/proofs. The results appear to be sound.

**Weaknesses:**

One notable weakness is that the central definition of the paper (Definition 3.1), albeit quite natural, is not entirely uncontroversial: the fact that most coalitions are not core-blocking may not be sufficient for certain scenarios. That is, it is not clear whether the introduced notion of fractional core stability is an appropriate notion of stability in hedonic games. It would be good to see some more motivation and suitable applications for this definition, as well scenarios wherein this notion fails to provide meaningful guarantees.

Besides the issue above, another weakness is that most of the results follow rather directly from prior work and standard techniques, but I don't view this as a basis for rejection.

**Questions:**

Some minor issues:

1. Technically speaking, the term $2^{-n^{1/3}}$ (for example in Theorem 5.1) is sub exponential, so maybe it is worth rephrasing the claims that $\epsilon$ is decreasing exponentially with the number of players (but I leave this up to the authors).
2. There is something off with the citation style; for example, Line 89 reads "Donahue u. Kleinberg" instead of "Donahue and Kleinberg." This also occurs in many places.
3. There are missing punctuation marks in some equations throughout the paper.
4. Regarding the title, maybe it is worth considering either removing the word "epsilon" or replacing "epsilon" with $\epsilon$; currently it reads slightly weird.

**Limitations:**

The authors have adequately addressed the limitations.

---

> ### Author Rebuttal · Authors · 2023-08-08
>
> We thank the reviewer for carefully reading our paper and for the constructive remarks. In the following we will address the comments in the 'Weaknesses' and 'Questions' sections.
>
> **Limitations of the definition:** A possible scenario where the notion of $\varepsilon$-FC stability shows its limitations are instances with a small number of agents $n$. In fact, in such cases, it is often easy to exhibit eventual core-blocking coalitions. However. as $n$ grows, finding such coalitions becomes computationally too hard, as shown by the many complexity results in the HGs literature. Thus, with a big enough $n$, one must think carefully about how such a coalition could be identified and how suitably degrees of stability can anyway be guaranteed. On this respect, the distribution $\mathcal{D}$ serves the purpose of modeling the probability of a coalition forming naturally (i.e. of a subset of agents meeting together) and eventually blocking.
>
> However, it is crucial to observe that every possible approximate core-stability concept must allow a small fraction of blocking coalitions to exist. Previously introduced notions (like the ones described in [17, 22, 26]) limit the number of allowable coalitions by imposing extra rules to the agents deviating from a certain configuration. In general, in all such relaxations, core-blocking coalitions are still there, but they are filtered out by the corresponding requirements. Since this approach seems unnatural in modeling scenarios in which agents are free to deviate anywhere they like, allowing only a fraction of all possible coalitions to core-block seems the most natural choice.
> As a concrete example to the above argument, consider the classical approximate core-stability concept in which a core-blocking coalition $C$ is acceptable if and only if the agents deviating to $C$ improve by at least a certain (additive or multiplicative) factor. As also pointed out in the scientific literature, this concept appears fragile, as there is no reason why even a subtle improvement should not be pursued by the agents.
>
> **Originality of the work:** Our algorithms are original, contain novel ideas, and are not based in any way on existing literature in HGs. Moreover, even if Chernoff bounds are a standard tool in probabilistic analysis and the PAC framework, we used them in an arguably original and non-trivial way, at least for what concerns the setting of HGs.
>
> **Minor issues:** We thank again the reviewer for pointing out to us the minor issues. We will address all of them in the final version of the paper.

---

> > ### Comment · Reviewer_psiE · 2023-08-10
> >
> > I thank the authors for the detailed response.

---

### Official Review · Reviewer_ot5e · 2023-07-06

**Soundness:** 3 good
**Presentation:** 2 fair
**Contribution:** 3 good
**Rating:** 7
**Confidence:** 3

**Summary:**

The manuscript at hand proposes a novel relaxation of core stability in coalition formation games, called eps-fractional core stability, which can be viewed as a requirement that a coalition selected uniformly at random from the set of all coalitions (conceptually choices according to some other distribution are also permitted, but the results in the submission are all for uniform selection) witnesses classic core instability with probability at most eps. While this is a relaxation of core stability, just like core stability, it cannot always be achieved, even for uniform distributions, fractional hedonic games and anonymous hedonic games. These impossibility results hold for a constant choice of eps, but on the other hand in these settings, the submission shows that eps-fractional core stable partitions exists for eps growing exponentially in the number of agents.

**Strengths:**

The motivating criticism of previous relaxations of core stability is justified and the proposed model is natural from a certain probably-not-discovering-core-instability perspective.
The quality of the writeup is high and I did not find any technical errors.

**Weaknesses:**

I find that it would be justified to contextualise the newly introduced stability notion with respect to more classic ones. For example, from my understanding, eps-fractional core stable outcomes are not even necessarily individually rational which can be viewed as extremely unnatural and should at least be discussed and possibly even somewhat justified.
On a related note, I do not really see how the definition reflects the fact that `the probability that groups of agents improve meeting by chance is very low' (l 53). Why would the probability that a coalition meets be uniform in the number of all coalitions. Naively, I would expect something more like the probability of each agent being included in a coalition be i.i.d. with some probability, meaning that small coalitions would be much more likely to meet. I think it would be good it the uniform selection of coalitions from the set of all coalitions could be motivated a bit more.

In terms of presentation, I do not understand why the impossibility results are stated as existence of some distribution results. Would it not strengthen the statements if one would simply say that D is the uniform distribution?

The authors satisfactorily addressed all weaknesses in the rebuttal period and I have changed my score to reflect this.

**Questions:**

Can you address my concerns above about the naturalness of eps-fractional core stability and the motivation of uniformly selecting coalitions from the set of all coalitions?

**Limitations:**

No concerns.

---

> ### Author Rebuttal · Authors · 2023-08-08
>
> We thank the reviewer for the insightful comments. We reply here to the points raised in the 'Weaknesses' and explain how we will address them in our paper. Before providing our detailed answers, we want to highlight that while for simple FHGs our results hold only for the uniform distribution, for anonymous HGs they apply to any $\lambda$-bounded distribution. Moreover, we believe that in the first case it would be possible to generalize to $\lambda$-bounded, even if at the moment this remains an open question.
>
> **Individual rationality:**
> Despite our main focus was relaxing the demanding notion of core stability, the reviewer raised an interesting point. It is indeed correct to claim that our relaxation does not imply individual rationality (IR). However, in the case of simple fractional HGs, any outcome is IR and therefore it is our proposed solution. In the case of anonymous HG, our algorithm provides an IR solution only if coalitions of size $1$ are less preferred than any other coalition size. That said, we believe it is possible to modify our algorithm, under the reasonable assumption that the value of the singleton coalition is known for each agent$^1$, in a way that also IR is satisfied. However, such a requirement may have an impact on the quality of $\varepsilon$. It would be definitely of interest to come up with such an algorithm and understand what impact the IR requirement would have on $\varepsilon$. We will include a discussion on IR in our paper and propose the study of $\varepsilon$-FC solution in conjunction with IR as an important open question.
>
> $^1$ Such an assumption has also been made in [18] to obtain PAC-stabilizability for the class of bottom responsive HGs. Moreover, it can be assumed that agents have been asked to express their valuations in a normalized fashion, that is, they value $0$ their singleton coalition.
>
>
> **Impossibility results:** The distribution $\mathcal{D}$ can be interpreted as the probability that a group of agents can interact/meet. Despite there exist more reasonable distributions than others, the distribution $\mathcal{D}$ may be in principle any. Therefore, with our impossibility results, we show that whenever the support of the distribution $\mathcal{D}$ is not the set of all possible coalitions, the guarantee on $\varepsilon$ may be very bad. For this reason, we focus on specific classes where the aforementioned condition is not satisfied. Clearly, any distribution whose support is the set of all possible coalitions is a $\lambda$-bounded distribution (since any coalition has a positive probability to be sampled); however, $\lambda$ could be an extremely high number.
>
> **Uniform, $\lambda$-bounded, and i.i.d. distributions:** As just mentioned, our choice of studying uniform and $\lambda$-bounded distributions follows by the provided negative results. We agree that the idea of having a probability distribution such that the probability of each agent being included in a coalition is i.i.d.\ with some probability $\alpha \in (0,1)$ is natural. Such a distribution is indeed a $\lambda$-bounded distribution as the support is the set of all possible coalitions. Therefore, our results for $\lambda$-bounded distributions apply also to this class. Being such a class particularly meaningful, we will point this out in our paper. Moreover, having this class a nice probabilistic structure, it would be interesting to understand if it is possible to improve the bound on $\varepsilon$ especially when the corresponding parameter $\lambda$ is extremely high, also exploiting the independence of the membership of each single agent.
>
> On a final note, despite its ability to properly represent the probability of agents meeting may be questionable, finding an $\varepsilon$-FC for the uniform distribution is equivalent to the problem of determining a coalition structure having the fraction of possible core-blocking coalitions bounded by $\varepsilon$. Providing a coalition structure having only a limited number of core-blocking coalitions seems a good compromise in the case in which no core stable outcome exists.

---

> > ### Comment · Reviewer_ot5e · 2023-08-16
> >
> > Thank you for your careful and detailed response. The promised discussion on the relationship to and possibility of including IR sufficiently address this point of concern for me. The response about the statement of the impossibility results also completely alleviate what I now think was just me being confused. Regarding the last point raised by me about the naturalness of the distributions, I appreciate the response of the authors and the fact that the iid inclusion of agents in possible blocking coalitions will be pointed out in the paper. I still disagree with the sentence including `the probability that groups of agents improve meeting by chance is very low' (l 53 of the original submission) and encourage the authors to change its phrasing, possibly in connection with the added remark about iid inclusion of agents in blocking coalitions. However, I trust that this is something that the authors will address appropriately and therefore revise my score to recommend acceptance.

---

### Official Review · Reviewer_UFzk · 2023-07-07

**Soundness:** 3 good
**Presentation:** 3 good
**Contribution:** 3 good
**Rating:** 5
**Confidence:** 3

**Summary:**

This paper introduces and studies the concept of epsilon-fractional core stability in Hedonic Games (HGs), which is a natural relaxation of core stability where a small ratio (or a small probability mass) of coalitions is allowed to core-block. The paper shows that this notion is very flexible and resilient to possible uncertainty of agents’ preferences.

The paper provides positive results for two fundamental classes of HGs under reasonable assumptions on the considered distributions. Specifically, the paper designs efficient algorithms returning an epsilon-fractional core-stable partition, with epsilon exponentially decreasing in the number of agents, for two classes of HGs: Simple Fractional and Anonymous.

On the negative side, the authors also prove that by allowing arbitrary sampling distributions an epsilon -FC may fail to exist for constant values of epsilon.

As discussed in the paper, the concept of epsilon-fractional core stability may stipulate subsequent work, including extending the positive results for simple fractional HGs to the more general class of lambda-bounded distributions, investigating the best guarantee for general distributions, and exploring other HG classes that have not been considered in the paper.

**Strengths:**

+ The new concept of epsilon-fractional core stability is a natural relaxation of core stability, which deserves to be studied as core-stable partitions seldom exist ( and even when they do it is often computationally intractable to find one).

**Weaknesses:**

- The Introduction of the paper immediately dives into technical details, which may overwhelm readers.

- There are clear gaps in the results. It will be much better if the results for simple fractional HGs to the more general class of lambda-bounded distributions. As the first work on of epsilon-fractional core stability, I do not view this as a serious drawback.

**Questions:**

Extending the results for simple fractional HGs to lambda-bounded distributions is not a trivial task. Can the authors briefly discuss the difficulties here?

The authors answered the question in detail in the rebuttal. Thanks!

---

> ### Author Rebuttal · Authors · 2023-08-08
>
> We thank the reviewer for the positive comments and constructive remarks. In the following, we will address the posed questions.
>
> As we discussed in the paper, $\lambda$-bounded distributions are a very general class. Lemma 3.4 outlines what we can essentially say about the probability of a certain family of events occurring for any $\lambda$-bounded distribution $\mathcal{D}$. When studying random variables of interest, this property and a smart use of known bounds can only help us to obtain confidence intervals on the size of sampled coalitions (see Lemma 6.3). In the anonymous case, the only random variable of interest is the $X$ modeling the size of a sampled coalition, which eased the calculations. Even in this somehow simpler case anyway, estimating the average of $X$ is a crucial step to prove the validity of the algorithm.
>
> On the other hand, simple FHGs are intrinsically more complex and the algorithm that we built reflects this complexity. In particular, the proof of Lemma 5.5 utilizes $X$ (as defined before) and an extra random variable say $Z$, modeling the number of friends of an agent contained in a sampled coalition. It might be the case that, with a much more elaborate proof, the procedure and the result can be adapted to the $\lambda$-bounded case. This would require first proving results allowing us to handle the interplay between $X$ and $Z$ and, in addition, carefully revisiting all the parameters involved.
>
> As the reviewer pointed out, being this work the first on $\varepsilon$-FC stability, our main aim was to show that positive results were indeed possible in a scenario dominated by powerful negative results. We believe that Theorem 5.1 is relevant even under this restricted assumption and might foster further research.

---

> > ### Comment · Reviewer_UFzk · 2023-08-16
> >
> > Thank you for the helpful response. I do not have additional questions.

---

### Official Review · Reviewer_eoec · 2023-07-07

**Soundness:** 3 good
**Presentation:** 3 good
**Contribution:** 3 good
**Rating:** 5
**Confidence:** 2

**Summary:**

This paper studies the problem of finding a core-stable partition in the hedonic games. As core-stable partitions seldom exist and they are usually hard to compute, the authors propose a relaxed notion of epsilon-fractional core-stable partitions, in which at most epsilon-fraction of all possible coalitions can core-block the partition. The authors show that the existence of epsilon-fractional core-stable partition depends on the distribution of sampling partitions that are allowed to core-block using the notion of lambda-bounded distributions. With constant lambdas, the authors design efficient algorithms to compute a epsilon-fractional core-stable partition.

**Strengths:**

1. This paper is generally well-written and clear. The problem studied in this paper is interesting and relevant to the algorithmic game theory community.

2. This paper provides an almost complete picture of the problem being studied. The results are non-trivial and technically strong.

**Weaknesses:**

1. It is unclear whether Algorithm 1 can only be applied to uniform distributions.

**Questions:**

1. What is the dependence on lambda in Theorem 5.1? Does it only apply to uniform distributions, i.e., when lambda = 1?

**Limitations:**

The authors adequately addressed the limitations.

---

> ### Author Rebuttal · Authors · 2023-08-08
>
> We thank the reviewer for the helpful comments. In the following, we will address the comment in 'Weaknesses' and the posed questions:
>
> In the case of simple FHGs (i.e. Section 5), we assume that the distribution is uniform, thus Theorem 5.1 works under this assumption. This shows that the definition of $\varepsilon$-FC is meaningful, as it is indeed possible to reach positive results in cases in which they could not be previously obtained without our relaxed requirements on the standard core-stability notion. We believe that it is possible to extend Theorem 5.1 to $\lambda$-bounded distributions without significantly modifying Algorithm 1. Presumably, this generalization would influence the guarantee on $\varepsilon$, which we expect to become parameterized in $\lambda$ as in the case of anonymous HGs. However, such an extension is technically quite challenging and at the moment constitutes an interesting open question. The reviewer might find also interesting our answer to reviewer UFzk who shared similar concerns.

---

> > ### Comment · Reviewer_eoec · 2023-08-18
> >
> > Thank you for the detailed response! I don't have other questions.

---

### Official Review · Reviewer_1m1t · 2023-07-27

**Soundness:** 4 excellent
**Presentation:** 4 excellent
**Contribution:** 3 good
**Rating:** 7
**Confidence:** 3

**Summary:**

This paper is a theory paper analyzing the stability characteristics of the Hedonic Game (HGs). HGs are a class of problems that partition agents considering their preferences. The core stability is the most important characteristic, which measures how well the agents' preferences are fulfilled. Specifically, this paper analyzes the \epsilon-fractional core (\epsilon-FC) stability, which means less than the coalitions of a ratio \epsilon is core-blocked, i.e., the preference was not fulfilled. In short, the paper analyzes famous classes of HGs, simple fractional and anonymous HGs, and shows that we can find \epsilon-FC solutions if \epsilon is large enough under \lambda-bounded distribution, which bounds the size of coalitions so that they are not too unbalanced. The paper also develops algorithms to find the \epsilon-FC solutions for those HG classes.

**Strengths:**

+ Though I am not familiar with HGs, the definition of \epsilon-FC and the settings of simple-fractional and anonymous HGs seems to be natural to many practical scenarios. Therefore, I consider this work to be useful for not only theoretical aspects but also practical applications deemed as HGs.
+ The theory part is well written, which can be easily understood by readers outside HGs (or even outside game theory, like me; I personally enjoyed reading the paper.)

**Weaknesses:**

- As described in the intros and related work sections, the \epsilon-FC stability is quite similar to PAC stability. [27] has shown PAC stability under \lambda-bounded distribution but in different games (W-games). Since I consider the theoretical aspect regarding the analysis of \epsilon-FC may be similar for many contexts, the theoretical contribution of this paper may be falling in a specific part, in other words, extending the work [27] to different games (i.e., HGs).
- Since the existence of \epsilon-FC solutions is bounded by an equation using \epsilon and \lambda relying on the number of agents, in practical scenarios, it might be difficult to find \epsilon-FC for some realistic settings.

**Questions:**

Q. Theoretical difference from [27]. Specifically, do we require notably different derivations and/or algorithms between PAC stability for W-games and \epsilon-FC for HGs?

Q. Can we find several practical examples in which cases (i.e., combinations of \lambda, \epsilon, and the number of agents) we can find \epsilon-FC-stabilized partitions? This would be quite useful to understand the usability of the proposed method intuitively.


**Limitations:**

For the theoretical aspect, I did not find notable limitations.

---

> ### Author Rebuttal · Authors · 2023-08-08
>
> We thank the reviewer for the positive comments and the useful feedback.
>
> To answer effectively to the reviewer's questions, we would like to focus on a couple of points that appear to be unclear. We did not understand exactly if the reviewer here wanted to quote paper [18] instead of [27]. In any case, [27] initiated the study of PAC learnability/stabilizability of HGs classes. In that work, the two subclasses of HGs, namely FHGs and $\mathcal{W}$-games, are shown not to be PAC stabilizable. [18] continues this line of work and obtains more general results. In particular, the authors first propose the use of $\lambda$-bounded distributions in the context of $\mathcal{W}$-games, which are computationally easier to analyze w.r.t. FHGs and Anonymous games.
>
> The definition and ideas in [18] were an inspiration for us, which is why we mention it in the paper. In that work, however, the authors are able to show a much stronger condition on the provided solution. Namely, regardless of the considered distribution, $\varepsilon$ can be arbitrarily close to 0. However, such a strong property is not always attainable, because of the freedom of both the distribution and the $\varepsilon$ parameter. For this reason, to deal with these negative cases, our work is a first attempt of relaxing such strong conditions still achieving good stability guarantees.
>
> In the following, we will answer the reviewer's questions in the order:
>
> 1) The algorithms are very different and strongly depend on the classes considered. We believe that they are interesting on their own, as they are based on novel ideas and the analysis is not trivial. The algorithm in [18] is a greedy procedure that also uses existent algorithms for certain values of $\varepsilon$ and the analysis is nice but, to our opinion, simpler. Our algorithms are completely different and the probabilistic analysis is more challenging from a technical perspective.
>
> 2) For any possible values of $\lambda$ and $\varepsilon$, when $n$ is small it is easy to search for blocking coalitions and figure out the existence of $\varepsilon$-FC partitions. As $n$ grows beyond a certain threshold, we show that such partitions always exist and can be efficiently computed, while conversely the standard problem of computing a core-stable solution is hard, even beyond NP. Thus, from a practical point of view, our results give effective ways to handle large HGs instances.
> As long as $\lambda$ is small with respect to $n$, we consider the bounds on $\varepsilon$ to be pretty powerful for any instance with many agents. Clearly, with a small $n$ it could be easy to search for blocking coalitions and an elaborate concept like $\varepsilon$-FC stability would not retain much practical sense. Anyway, we believe that the flexibility allowed by the dependence on $\lambda$ is still of interest in applications where the sampling distribution is somehow more exotic.

---

> > ### Comment · Reviewer_1m1t · 2023-08-17
> >
> > Thanks for the detailed responses!

---

### Author Rebuttal · Authors · 2023-08-08

We thank the reviewers for their valuable comments. We will take into account their feedback and revisit the paper accordingly. In what follows, we provide a general answer to the reviewers addressing some shared concerns raised in their comments.
The specific questions will be addressed in the dedicated responses to the reviewers. We will be happy to reply to any further questions in the discussion phase.

The idea underlying the definition of $\varepsilon$-fractional core-stability is that a partition that can be core-blocked by at most an $\varepsilon$ fraction of all possible coalitions is a good candidate to approximate a core-stable solution in the classical sense. From a probabilistic viewpoint, this is equivalent to asking that a coalition sampled uniformly at random has a probability of core-blocking the proposed partition upper bounded by $\varepsilon$. Therefore, the most natural choice and the very starting point of our reasoning is the uniform distribution, which is why we consider studying it meaningful and not restricting as meant in this respect.

The probabilistic interpretation led us to generalize the definition to incorporate any possible sampling distribution $\mathcal{D}$.
As a further step in this direction, this allowed us also to extend our study to all the situations in which preferences are unknown and must be learned by sampling. In such a setting, the sampling distribution $\mathcal{D}$ can also be interpreted as the probability that a group of agents spontaneously meet in the considered case study, and can be used to both learn the agents' utilities and verify the stability of the created partition.

Unfortunately, the results in Section 4 show that not all distributions $\mathcal{D}$ allow positive results: in particular a condition that appears to be crucial is that the support of $\mathcal{D}$ contains all coalitions. We thus decided to focus our attention on the class of $\lambda$-bounded distributions, which is the largest class of distributions having support on the set of all possible coalitions. First introduced in [7] as a reasonable choice for the PAC learning setting and then used in [18] in the context of PAC-stabilizability, it is a general enough class with interesting properties that seems to fit very well in this scenario.

In our view, this work is a first step in understanding which kinds of coalition structures are worthy of being considered stable in real-world applications where the classic definition of core stability might be too demanding and/or hard to compute.

---

### Decision · Program_Chairs · 2023-09-21

**Decision:**

Accept (poster)

**Comment:**

This paper got rather positive reviews, and its scores (7,7,6,5,5) put it clearly in the borderline/accept candidate category. Specifically, when asked to chime into the accept vs reject decision discussion, both of the positive reviewers repeated their support for an accept decision. Overall, I see a solid paper that introduces a natural relaxed notion of core-stability (which is inspired by PAC-stabilizability in [27]). This notion although not free of potential criticism, seems to be a natural one to explore. The paper does a thorough job in understanding its promises and limitations of this notion (by exploring existence and computational aspects for different classes of distributions and different classes of games).